# Evaluation of the industrial cooperation of the Guangdong Hong Kong Macao Greater Bay Area: Based on the Origin-Destination pairs of industrial parks and coupling model efficiency

**Sa Ma**[☉], **Jinge Ding**[☉], **Zhengdong Huang**＊, **Renzhong Guo**

Research Institute for Smart Cities, School of Architecture and Urban Planning, Shenzhen University, Shenzhen, PR China

[☉] These authors contributed equally to this work.

＊ zdhuang@szu.edu.cn

**Data Availability Statement:** The POI and AOI data of industrial parks are available from the Amap official interface (https://lbs.amap.com/api/

## Abstract

The industrial cooperation of Guangdong Hong Kong Macao Greater Bay Area (GBA) is one of the leading regional development strategies of this world-class urban agglomeration. This study constructed the industrial cooperation network based on the travel Origin-Destination (OD) connections among industrial parks. A multi-dimensional industrial cooperation and industrial development calculation index system were also set up to measure the nonlinear interaction relationship between them. The research found that an industrial collaboration network has been basically formed in the GBA, particularly presented by major cities. Some undeveloped cities may receive more benefits in the industrial collaboration network. The Covid-19 pandemic has had an impact in terms of within city connection instead of cross-city industrial cooperation. In addition, the degree of coupling between urban industrial coordination and urban industrial development has improved significantly over one decade and taking industrial collaboration as the input variables, industrial collaboration efficiently leads to industrial development outputs in almost every city in the GBA. Practically, decision makers should encourage and support intercity industrial collaboration, particularly between cities with closer geographic proximity, as it has been found to result in stronger cooperation and better economic enhancement. In addition, although industrial collaboration does not guarantee industrial development, when the collaboration systems and policies are enhanced, the synergy and coordination between them gradually improve. This highlights the potential benefits of continued investment in industrial collaboration for economic development.

## Introduction

The global economy is currently facing numerous uncertainties and challenges, especially in the wake of the Covid-19 outbreak, including trade protectionism, supply chain

webservice/guide/api/search). The city-level GDP data can be found on the special page on Statistics of the GBA (https://www.dsec.gov.mo/BayArea/zh-CN/#s5). The data for the coupling model came from China's economic and social big data research platform (https://data.cnki.net). Amap navigation data are available from the GitHub database (https://github.com/Ding74/PLOSONE-MANUSCRIPT-D2304060).

**Funding:** Our research continues to be supported by the National Key Research and Development Program: (Project No: 2019YFB2103100); the Guangdong Science and Technology Strategic Innovation Fund (the Guangdong–Hong Kong-Macau Joint Laboratory Program, Project No.: 2020B1212030009); Shenzhen Key Laboratory of Digital Twin Technologies for Cities (Project No.: ZDSYS20210623101800001). The funder of project No. 2019YFB2103100 purchased Amap navigation data. Other funders had no role in study design, data collection and analysis, decision to publish, or preparation of the manuscript.

**Competing interests:** We reiterate that there are no competing interests associated with this study, a fact that has been duly disclosed in the article.

fragmentation, and sluggish global economic growth. As a result, industrial chains and collaborations between cities and regions are undergoing unprecedented changes. Developed countries are reshoring some low-end segments of their industrial chains to ensure employment and stable economic growth, while developing countries are facing increased competition. The study of industrial collaboration is a central aspect of strategic management [1]. "Cooperation is becoming increasingly important in the modern business environment" [2]. As mentioned by Child et al [3], Eastern countries have an advantage over Western countries in terms of collective actions and cooperation. Additionally, governments consider industrial collaboration as a major driving force for regional development. In China, coordinated regional development has become a strategic priority for promoting steady economic growth and addressing economic uncertainties [4]. In 2017, the Chinese government identified coordinated regional development as a leading regional development strategy.

Clemons and Row [2] pointed out that cooperation aims to increase resource utilization and value through higher explicit coordination of economic activities. Babkin et al [5] argue that cooperation is an objective phenomenon as individual economic units are not as effective as outsourcing, industrial clusters, and strategic alliances, which serve as models for cooperative development. Industrial cooperation refers to the complementary and collaborative use of resources, information, and technology among different enterprises and industries within different geographic regions to achieve more efficient and competitive production and innovation [6]. Its theoretical foundation is based on Marshall's external economy theory [7, 8]. From the perspective of urban economics, industrial cooperation in mega cities reflects the significance of diversified industrial structures and ensures the stable economic growth of major cities [9, 10]. Industrial cooperation also aims to improve economic efficiency [11] and alter regional industrial layouts [12, 13]. For instance, industrial cooperation can optimize spatial equity by increasing density, reducing distance, and minimizing segmentation [14, 15]. Furthermore, the strengthening of industrial cooperation between cities leads to the integration and optimization of industrial chains, the formation of new industrial clusters, and the improvement of regional industrial added value and economic benefits [16]. As a representative of developing countries, research on China's industrial coordinated development provides important references for other developing countries in addressing regional development imbalances.

Despite the widespread recognition of the concept of industrial cooperation and ongoing policy refinements, research on industrial cooperation lags behind strategic deployment and policy implementation. Firstly, in terms of measuring industrial cooperation, qualitative research methods such as case studies [17, 18] and interviews [19] are commonly used, while econometric methods that can quantitatively analyze the collaborative relationship between enterprises often lack statistical data across administrative boundaries. Secondly, although industrial cooperation is considered an important means of improving industrial development, for instance, the influence of cooperative strategy on technology development [20], firm performance [21, 22], and economic development efficiency [23], insufficient research has been conducted on the coupling relationship between industrial cooperation and intercity economic development. This includes the dynamic relationship between industrial cooperation and economic development and how economic systems dynamically match with industrial cooperation, moving from a state of discord to coordination.

The Guangdong Hong Kong Macao Greater Bay Area (GBA) is a world-class urban agglomeration comprising nine cities in Guangdong province, Hong Kong, and Macao. It serves as a demonstration zone for deep cooperation between Mainland China, Hong Kong, Macao, and the scientific and technological innovation center. With a permanent population of 86.1719 million as of December 2020, the GBA is a crucial region for the development of

regional cooperation and industrial planning (The Outline of the Development Plan for the GBA). Industrial parks play a significant role as physical hubs for industry development and key nodes within the network of industrial collaboration [24–29]. The connections between industrial parks are essential for evaluating regional cooperation, reflecting the characteristics of the industrial spatial structure and calculating the intensity of industrial spatial distribution [30–33].

In this study, we focus on the GBA as a case study and utilize abundant travel navigation big data to construct a method for evaluating industrial cooperation based on Origin-Destination (OD) pairs, aiming to update research methods in this field. The concept of OD pairs is commonly used in the transportation field to describe the flow or demand of people, vehicles, and other transportation modes from an origin to a destination. Typically, this concept is utilized for traffic demand prediction and urban planning, and it is studied and analyzed using mathematical models and transportation data analysis methods. In our study, we utilized data from the two-year period of 2019 and 2020 in the GBA, provided by Gaode Company, and defined the traffic connection between two industrial parks, with one serving as the origin and the other as the destination, as an OD pair. A total of 995,893 OD pairs were established. Additionally, this study also examines the coupling relationship between non-spatial elements of industrial collaboration and industrial development. The aim is to update research methods and develop a fundamental understanding of how industrial collaboration influences industrial development.

## Theoretical background

The basic theory of industrial cooperation, or synergy, originates from the synergetics pointed out by Igor Ansoff in 1965 and it emphasizes a symbiotic and mutually reinforcing relationship between enterprises [34]. On this basis the German physicist H. Haken put forward the concept of synergetics in 1971. He believed that synergy refers to the behavioral process of joint development achieved by mutual promotion and coordination among various organizational parts within the system. Through the cooperative process of subsystems in an open system, the system formed new orderly structures and functions [35, 36]. In the regional studies the cooperation development of cities refers to urban agglomeration as an open system far from equilibrium, through the exchange of outside material or energy, and the cities form an orderly structure in time, space and function [37]. From the perspective of exogenous policy intervention, due to the existence of space costs, the economy development could not have a perfect competitive equilibrium; thus the regional coordinated development refers to the intervention mechanism adopted by the government for regional economic development in order to correct the "space failure" of the market [38]. Chen [39] emphasized the strategic role of cooperation development in enabling backward regions to catch up with others and narrow the development gap among cities, highlighting the importance of coordinated development and integration. In this process the core cities need to perform subtraction to ease some of their core functions and promote the moving out or transfer of their industries to relatively undeveloped areas. In this view industrial cooperation is the motive power of regional development. However, the mechanism underlying the coupling relationship between industrial cooperation and regional industrial development remains unknown.

In terms of industrial cooperation evaluation methods there are mainly three: Firstly, based on regional economic statistics data an evaluation index system is formed. For example, Zeng [40], Zeng, Cao [41], and Zeng, Yang [42] built different evaluation index systems covering economy, technology, infrastructure and ecology to measure the collaborative development capacity of the Yangtze River Delta cities. Zhu and He [43] built an evaluation index system

consisting of a development indicator, a cooperative indicator and an ecological civilization, population development and enterprise development indicator to evaluate the level of the Beijing Tianjin Hebei industrial cooperation. Chen and Li [44] analysed the level of the Beijing Tianjin Hebei regional industrial coordination through the per capita GDP, annual GDP and GDP growth rate and the market integration level. Secondly, industrial network models are built based on the input-output table, including the Campbell model, MFA model, ICA model, ECA model, etc., which reflect the regional industrial cooperation by the changes to, and transfers of, the industrial structure of the region, the location entropy of the industrial specialization level, the scale of the industrial agglomeration and the industrial concentration coefficient, etc. [45, 46]. For instance, He, Pan [47] constructed forward and backward industrial linkages of foreign-funded enterprises through the Zhejiang input-output table to study the synergy between foreign and domestic enterprises. Thirdly, many scholars use spatial factors to evaluate industrial synergy, such as Qin, Zhang [48], Qin and Cui [49], who constructed an evaluation method for coordinated development of the regional economy through Moran's I index and economic growth rate and the increasing level variable coefficient, or by seeking a synergistic agglomeration relationship between two industries, such as producer services and manufacturing [7, 50–54]. However, two main issues of industrial cooperation have been given more attention: the regional industrial cooperation cannot be decided without constructing a network of regional industries and an evaluation of industrial cooperation cannot override economic development. That is how industrial cooperation and economic development have interaction influences and how industrial cooperation inputs lead to economic development.

In this study the social network theory and coupling theory provide a meaningful way to present the forming of a regional industries connection and the relationship between industrial cooperation and economic development. The social network theory emphasizes the role of enterprise cooperation and the influence of embeddedness and network relationship on enterprise geographical agglomeration [55, 56]. The California School, represented by Scott and Storper, pays attention to the research on the interaction between regional internal linkages and production relations [57], which becomes a widely used method to form the connections among actors such as firms and cities to evaluate the industrial collaboration status. Harrison [58] believed that, due to the need for interpersonal contact, a large number of interconnected small enterprises realized specialization in the production process, forming closed cooperation, sharing production equipment, information and skilled labor and constructing a closely related network, resulting in industrial agglomeration and upgrade. The individuals, enterprises and institutes are "embedded" within networks and extended, renewed and transitioned based on the connections with other actors [59, 60]. The coupling theory is a principle that describes the relationship between two or more systems through internal mechanisms, and has been applied to various disciplines [61]. The role of coupling and the degree of coordination between two systems reflect the degree of cooperation in development and evolution, thus showing the trend of system from disorder and disharmony to order and coordination [61]. The existing research has examined the coupling relationship among different factors of internal economic development in the system such as in urbanization and the environment [62]or urban economic development and air quality [63]or water environment [64], but only a few have paid attention to the industrial coordination and industrial development of urban agglomeration. Understanding the coupling relationship between industrial coordination and industrial development is valuable in gaining a deeper understanding of the essence of industrial coordination and is helpful for regional decision-makers to make suitable decisions on regional economic development.

## Research method

### Industrial parks OD connections and network analysis

At present there is no consolidated measurement index for industrial collaboration. This study formed the industrial cooperation network of the GBA according to travel navigation data (OD pairs) starting from any one of the industrial parks and ending at another industrial park based on AMaps App, which is China's largest provider of mobile digital maps, navigation and real-time traffic information, with 60 million active users per day. Firstly, the interface of the Amap was used to obtain the POI classified by the Amap official as an industrial park. The industrial parks mentioned in official government reports have also been additionally collected and a total of 8920 industrial parks were obtained.

Secondly, in order to avoid bias in the process of delimiting the boundaries of the industrial parks, the AOI (areas of interests) provided by AMaps were used to vectorize the boundaries of the industrial parks. For the missing and inaccurate boundary information, the study manually corrected according to the road boundaries, entrance of the industrial parks and the relationship between industrial parks and surrounding buildings. The travel navigation pairs with both origins or destinations located within industrial park boundaries are collected. The OD connections are classified as OD connection within cities and OD connection across cities in four time periods, January 2019—June 2019, July 2019—December 2019, January 2020—June 2020, and July 2020—December 2020, in order to avoid bias caused by the Covid-19 epidemic situation. 417827 and 578066 OD pairs were formed in 2019 and 2020, respectively. Finally, the industries parks are classified according to the way of naming and key words selection. Restricted by the feasibility of data and government naming convention, two types of general industries park (general industrial park and general science and technology park) and six special industries parks (high-tech park, mass entrepreneurship and innovation park, logistics park, cultural industry park and e-commerce industrial park, biochemical and agricultural ecological park) are been defined according to the key words, which is commonly used in official institutions (such as the China Institute of Logistics Prospective Industry Research Institute). The key words in this article came from the name of the park and the registered business type (see Table 1).

### Industrial cooperation and industrial development relationship

Variables were normalized and weighted according to the Analytic Hierarchy Process (AHP). Through the improved coupling coordination model (CCD) and data envelopment analysis

**Table 1. The industrial park classification.**

| Classification | | |
|---|---|---|
| **General industries park** | General manufacturing park | |
| | General science and technology park | Science park, science and technology industry etc. and other high-technology parks that cannot be classified in the special industries parks below |
| **Special industries parks** | High-tech park | High-tech, high-technology, new technology, etc. |
| | Mass entrepreneurship and innovation park | Entrepreneurship, incubation, acceleration, innovation, innovative technology, creative arts, mass entrepreneurship, etc. |
| | Logistics park | Logistics, freight, freight transport, cold chain, storage, warehousing, bonded, under bond, etc. |
| | Cultural industry park | Culture, creativity, design, animation, jewelry, film and television, advertising, etc. |
| | E-commerce industrial park | E-commerce, commerce, e-commerce and trade |
| | Biochemical and agricultural ecological park | Biology, chemistry, petrochemical, materials, energy, health, genes, medicine, vaccines, aquatic products, food, ecology, agriculture, flowers and seedlings, farms, agricultural products, etc. |

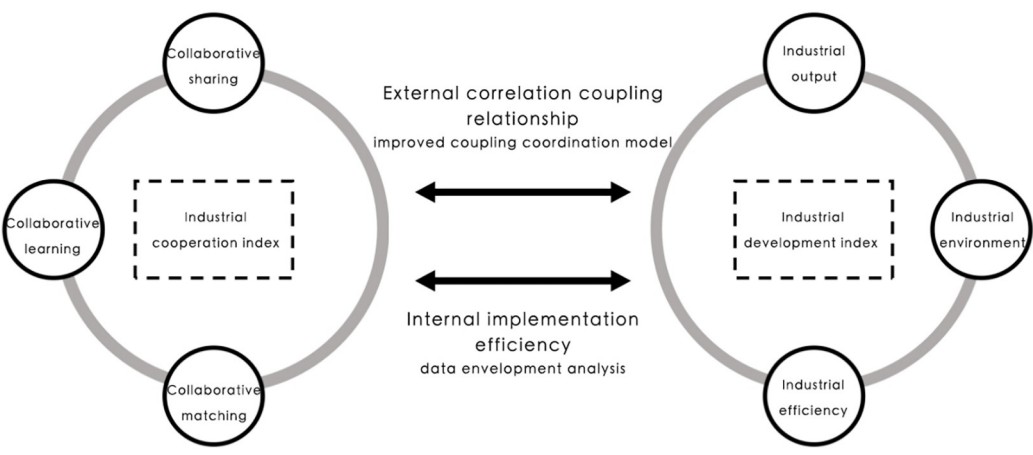

**Fig 1. Industrial cooperation and industrial development relationship sketch map.**

model (DEA), the external correlation between industrial collaboration and industrial development, as well as the internal investment efficiency, are introduced to measure the nonlinear interaction relationship between industrial coordination and industrial development (see Fig 1). The study set up the multi-dimensional industrial cooperation and industrial development calculation index system according to the three dimensions of industrial cooperation, namely collaborative sharing, collaborative learning and collaborative matching, and the three dimensions of industrial development, namely industrial output, industrial environment and industrial efficiency (see Table 2).

**External correlation coupling relationship between industrial cooperation and industrial development based on the improved coupling coordination degree evaluation.** The formula of the traditional coupling model is shown below.

$$D = \sqrt{C \times T} \tag{1}$$

$$C = 2\left\{\frac{U_1 U_2}{(U_1 + U_2) \times (U_1 + U_2)}\right\}^{1/2} \tag{2}$$

**Table 2. Industrial cooperation and industrial development calculation index system.**

| U1 | | | U2 | | |
|---|---|---|---|---|---|
| Destination layer | Criterion layer | Standard layer | Destination layer | Criterion layer | Standard layer |
| Collaborative sharing | Infrastructure sharing | Investment in fixed assets; Mileage of expressway | Industrial output | Product output | GPD growth rate; industrial added value |
| | Labor sharing | Average employees in high-tech industries; Current number of R&D personnel | | Innovation spillover | Number of patent applications; Number of patents granted |
| Collaborative learning | Administrative performance | No. of administrative licensing matters; No. of public service matters | Industrial environment | Manufacturer confidence | Purchasing manager index |
| | Innovation environment | R&D expenditure; Proportion of R&D expenditure in GDP | | Industrial development level | No. of enterprises above designated size; No. listed company |
| Collaborative matching | Market environment | Disposable income of urban residents | Industrial efficiency | Per capita efficiency | Labor productivity |
| | Investment matching | Actual utilized foreign capital | | Energy consumption reduction | Energy consumption reduction |

U1 is Industrial cooperation index.

U2 is Industrial development index.

$$T = \alpha U_1 + \beta U_2 \tag{3}$$

*C* means coupling degree function; the higher the *C* value, the higher the degree of coupling, and the value range is (0, 1). *D* refers to the coupling coordination degree. *T* is the comprehensive coordination index of the two systems of industrial coordination and industrial development, reflecting the contribution of the comprehensive development level of the two systems to the coupling coordination degree. $\alpha, \beta$ is an undetermined coefficient, which is usually defined subjectively. In many studies $\alpha = 0.5$ and $\beta = 0.5$.

According to the latest research results, in practice, in order to improve the coupling between the two systems, more efforts are needed for the underdeveloped system [61]. Therefore, less developed systems should assume higher weight values to achieve better coordination and attract more government attention to improve the performance of the underdeveloped system. Moreover, the higher weight values should be increased along with the increasing gap between the two systems.

In this study the values of $\alpha$ and $\beta$ are shown below:

$$\alpha' = \frac{U_2}{U_1 + U_2} \tag{4}$$

$$\beta' = \frac{U_1}{U_1 + U_2} \tag{5}$$

$$T' = \alpha' U_1 + \beta' U_2 \tag{6}$$

$$D' = \sqrt{C \times T'} \tag{7}$$

**Internal implementation efficiency of industrial cooperation and industrial development based on data envelopment analysis (DEA).**   DEA was originally proposed by Charnes, Cooper and Rhodes to evaluate the relative efficiency of the production system based on multiple inputs and outputs, namely the CCR model. On the basis of the CCR model, BCC model, CCGSS model and CCW model data envelopment analysis has been put forward successively. Tone [65] proposed a non-radial and non-oriented Slacks-Based Measure (SBM-DEA) model. Directional Distance Function (DDF) analysis is one of the most commonly used techniques in DEA, and allows the influence of perfect inputs and undesirable outputs to be considered respectively. Combined with SBM this study used the slack-based measured directional distance function (SBM-DDF) model mentioned by [66], which combines SBM with DDF and effectively avoids the radial characteristics and directionality of the DDF model and reduces the overestimation of efficiency. This study used MaxDEA7 Ultra software to process the SBM-DDF model to evaluate the input-output efficiency of industrial collaboration and industrial development quantitatively and discussed the internal interaction effect of industrial collaboration and industrial development.

## Result

### The industrial cooperation of GBA based on OD pairs

At the GBA level, in general, despite the influence of Covid-19, the OD pairs in 2020 (84077) were significantly higher than they were in 2019 (54206) (see Table 3), and even in the first

**Table 3. The network characteristics of industrial cooperation according to OD pairs.**

| | Network characteristics | | | | Degree centrality | |
|---|---|---|---|---|---|---|
| | Notes | Lines | OD pairs | Density | Out | In |
| **2019** | 9 | 65 | 54206 | 0.9028 | 0.1134 | 0.1174 |
| **2019 Jan-Jun** | 9 | 61 | 22927 | 0.8472 | 0.1128 | 0.1164 |
| **2019 Jul-Dec** | 9 | 65 | 31279 | 0.9028 | 0.1138 | 0.1182 |
| **2020** | 9 | 70 | 84077 | 0.9722 | 0.1153 | 0.1190 |
| **2020 Jan-Jun** | 9 | 68 | 33999 | 0.9444 | 0.1150 | 0.1181 |
| **2020 Jul-Dec** | 9 | 68 | 50078 | 0.9444 | 0.1154 | 0.1195 |

half of 2020, considered as having a stronger impact of the epidemic, the OD pairs were still more than 1000 times higher than in the January to June 2019 period. Relatively, the industrial cooperation networks in 2020 have more lines and a higher density than they did in 2019. The lines in 2020 and 2019 are 70 and 65, respectively, among all of the 72 (9*8) cities' pairs, which means more cities are involved in industrial park networks. The density of the industrial park networks of GBA was 0.9028 in 2019 and 0.944 in 2020. The GBA industrial parks have formed a very strong connection between cities. These connections are a little stronger in the second half-year than in the first.

According to Fig 2 the strongest connections occurred between the mega city Shenzhen and its integrated city Dongguan, followed by the mega city Guangdong and its integrated city Foshan. This reflects the cities with the highest GDP and the surrounding medium-developed adjacent cities have the highest connection. The industrial transfer of big cities has been formed. The cities with a higher GDP will play a more important role in the network, such as Dongguan, Shenzhen and Guangzhou. The connections between Shenzhen and Huizhou, Dongguan and Guangzhou, and Dongguan and Huizhou are also strong, which reflects, along with the economic development of Dongguan, it has taken more responsibility to contact

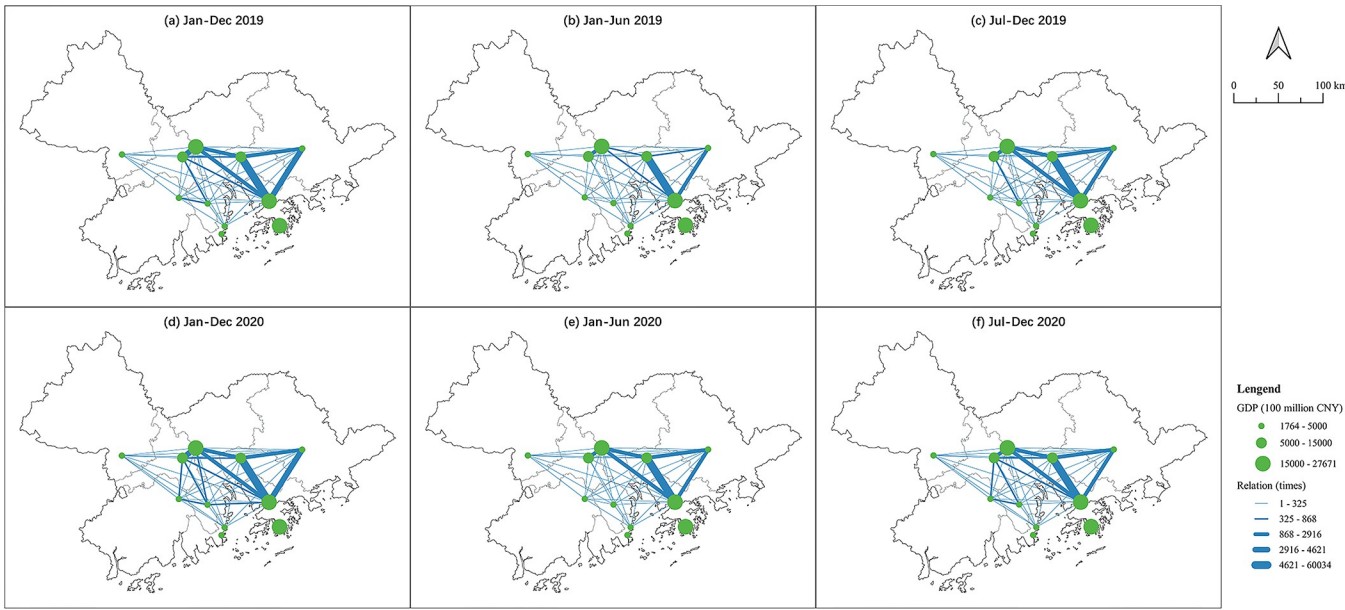

**Fig 2. The GBA industrial cooperation network according to OD pairs.** Reprinted background map from the National Catalogue Service for Geographic Information (www.webmap.cn) under a CC BY license, with permission from the Ministry of Natural Resources of China, original copyright 2020.

**Table 4. The centrality of the industrial cooperation network at city level according to OD pairs.**

| id | Degree centrality | | | | Closeness centrality | | between centrality |
|---|---|---|---|---|---|---|---|
| | Outdeg | Indeg | nOutdeg | nIndeg | OutClose | InClose | nBetweenness (%) |
| **2019** | | | | | | | |
| **Dongguan** | 22465 | 20443 | 0.138 | 0.125 | 1.000 | 0.889 | 1.369 |
| **Foshan** | 2512 | 2380 | 0.015 | 0.015 | 1.000 | 1.000 | 2.530 |
| **Guangzhou** | 3394 | 3700 | 0.021 | 0.023 | 1.000 | 1.000 | 2.530 |
| **Huizhou** | 2214 | 2655 | 0.014 | 0.016 | 0.889 | 0.800 | 0.000 |
| **Jiangmen** | 519 | 521 | 0.003 | 0.003 | 0.889 | 1.000 | 1.786 |
| **Shenzhen** | 21859 | 23049 | 0.134 | 0.141 | 1.000 | 1.000 | 2.530 |
| **Zhaoqing** | 72 | 97 | 0.000 | 0.001 | 0.727 | 0.727 | 0.000 |
| **Zhongshan** | 875 | 1042 | 0.005 | 0.006 | 0.889 | 1.000 | 1.458 |
| **Zhuhai** | 296 | 319 | 0.002 | 0.002 | 0.889 | 0.889 | 0.298 |
| **2020** | | | | | | | |
| **Dongguan** | 35810 | 30727 | 0.139 | 0.119 | 1.000 | 1.000 | 0.595 |
| **Foshan** | 3978 | 3695 | 0.015 | 0.014 | 1.000 | 1.000 | 0.595 |
| **Guangzhou** | 5504 | 6043 | 0.021 | 0.023 | 1.000 | 1.000 | 0.595 |
| **Huizhou** | 3607 | 3814 | 0.014 | 0.015 | 1.000 | 1.000 | 0.595 |
| **Jiangmen** | 972 | 934 | 0.004 | 0.004 | 1.000 | 1.000 | 0.595 |
| **Shenzhen** | 32273 | 36656 | 0.125 | 0.142 | 1.000 | 1.000 | 0.595 |
| **Zhaoqing** | 135 | 197 | 0.001 | 0.001 | 1.000 | 0.800 | 0.000 |
| **Zhongshan** | 1408 | 1623 | 0.005 | 0.006 | 0.889 | 1.000 | 0.000 |
| **Zhuhai** | 390 | 388 | 0.002 | 0.002 | 0.889 | 1.000 | 0.000 |

other cities. The cooperation between two big cities is at the medium level, which reflects the industrial division has been formed in these two cities. From 2019 to 2020 more cooperation was created and the network became more closely connected and more marginal cities, or economically backward cities, established connections. For instance, the OD pairs between Jiangmen and Foshan, Zhongshan and Guangdong, and Shenzhen and Zhongshan have increased significantly, and some new connections occurred in Zhaoqing, which has one of the lowest GDPs in the GBA (such as Huizhou and Zhaoqing, and Zhuhai and Zhaoqing).

At city level Shenzhen has the largest number of out and in industrial OD connections and the highest closeness centrality (see Table 4). As one of two big cities in the GBA and taking the main innovation and scientific research responsibility, compared with Guangzhou, Shenzhen has greater influence on creating industrial cooperation of the GBA. Guangzhou took more of a role as the center of culture and politics. Dongguan, which is considered as a manufacturing center and the backbone to undertake industrial transfer in the GBA, has one of largest number of OD pairs (22465 out and 20443 in) and the highest degree of centrality and Closeness centrality. This reflects that Dongguan also plays an essential role in the industrial cooperative network. The industrial transfer from Guangdong and Shenzhen to Dongguan has already been constructed. On the contrary, Zhaoqing, Zhuhai and Jiangmen have the smallest numbers of OD pairs in the network and they are all concentrated in the southwestern part of the GBA and have the lowest GDP in the region. The lack of industrial connections may result in a low economic development level, and Guangdong may have to take more responsibility to promote the economic development of these cities and form more connections with them. The standardizing degree centrality of these three cities rose from 0, 0.002 and 0.003 to 0.001, 0.002 and 0.004, respectively, from 2019 to 2020, which reflects that their roles in the network have increased. In terms of out and in OD pairs, two of nine cities have

**Table 5. The industrial connections within cities according to the OP pairs.**

| Inside cities | 2019 | 2019Q1 | 2019Q2 | 2019Q3 | 2019Q4 | 2020 | 2020Q1 | 2020Q2 | 2020Q3 | 2020Q4 | 2020–2019 | 2020-2019Q1 |
|---|---|---|---|---|---|---|---|---|---|---|---|---|
| Dongguan | 38713 | 6933 | 9461 | 10486 | 11833 | 56747 | 6338 | 15201 | 17103 | 18105 | 18034 | -595 |
| Zhongshan | 2224 | 395 | 561 | 551 | 717 | 3075 | 284 | 839 | 904 | 1048 | 851 | -111 |
| Foshan | 13300 | 2350 | 3213 | 3773 | 3964 | 17984 | 1853 | 4611 | 5598 | 5922 | 4684 | -497 |
| Guangzhou | 25571 | 4818 | 6061 | 6863 | 7829 | 37284 | 4119 | 9211 | 11696 | 12258 | 11713 | -699 |
| Huizhou | 2081 | 368 | 488 | 548 | 677 | 3330 | 303 | 878 | 1005 | 1144 | 1249 | -65 |
| Jiangmen | 774 | 138 | 187 | 213 | 236 | 1267 | 124 | 316 | 391 | 436 | 493 | -14 |
| Shenzhen | 279788 | 52770 | 67083 | 75230 | 84705 | 372702 | 38347 | 101146 | 115378 | 117831 | 92914 | -2000 |
| Zhuhai | 1074 | 201 | 243 | 284 | 346 | 1465 | 157 | 375 | 450 | 483 | 391 | -44 |
| Zhaoqing | 58 | 12 | 15 | 13 | 18 | 98 | 14 | 22 | 33 | 29 | 40 | 2 |

more out city OD pairs than in city OD pairs, and these two cities have endured more blame for production than consumption in the cities and provide strong power for the network.

In terms of inter city connections, in general, the OD pairs in all of the cities decreased in the 2020 first quarter when compared with 2019, which was caused by the pandemic (see Table 5). Although the barriers to traffic between cities were larger during the Covid-19 period, this did not have a great impact on the industrial links between cities, but had a greater impact within cities. The influence was soon weakened in the last quarter. At city level the links of industrial parks in Shenzhen were significantly higher than in other regions. This reflects that Shenzhen is the highland of the GBA industrial parks and there are many connections within the industrial parks. Guangzhou and Dongguan also have high inter city industrial connections (see Figs 3, 4), especially in Guangzhou, which does not have the same prominent position as Shenzhen and Dongguan in the industrial collaboration network among the GBA cities. This reflects that Guangdong has more potential to carry on with industrial transfer and drive development of the surrounding cities. Compared with the connections in the GBA industrial parks network, Zhaoqing and Huizhou have a smaller number of inter city links, reflecting that they received more benefits from the GBA industrial cooperation than they developed on their own. The other undeveloped cities, such as Jiangmen and Zhuhai, may try to closely participate in the industrial collaboration network to achieve more development opportunities.

At the micro-level, when we consider the industrial cooperation of the different types of industries in the GBA, it can be seen that the industrial division is obvious in the GBA (see Figs 5 and 6). The connection of high-tech parks is concentrated in Dongguan and Shenzhen and the high-tech parks also mainly belong to Shenzhen and Dongguan. This confirms the position of Shenzhen as the research and technology center and also reflects the significant role of Duangdong in high-technology development. Shenzhen and western Guangdong are the two main cultural industry centers in the GAB and spread to eastern Foshan and Dongguan, and both inter city and cross city connections are formed mainly in these four cities. In addition, the large percentage of mass entrepreneurship and innovation parks belong to Shenzhen and Guangdong while the connection of mass entrepreneurship and innovation parks are more concentrated in Dongguan and Shenzhen, especially within Dongguan and around Dongguan, and its surrounding areas, Foshan and even Zhuhai and Jiangmen. Most logistics parks belong to two megacities Shenzhen and Guangdong, which reflects the developed areas have more advantages in transportation system. The OD pairs of logistics parks have the largest range compared with other industry parks and cover edge cities like Zhaoqing, Zhongshan and Huizhou. The industrial cooperation of e-commerce and biochemical and agricultural ecological industrial parks mainly occurs within the three cities of Guangzhou, Dongguan and

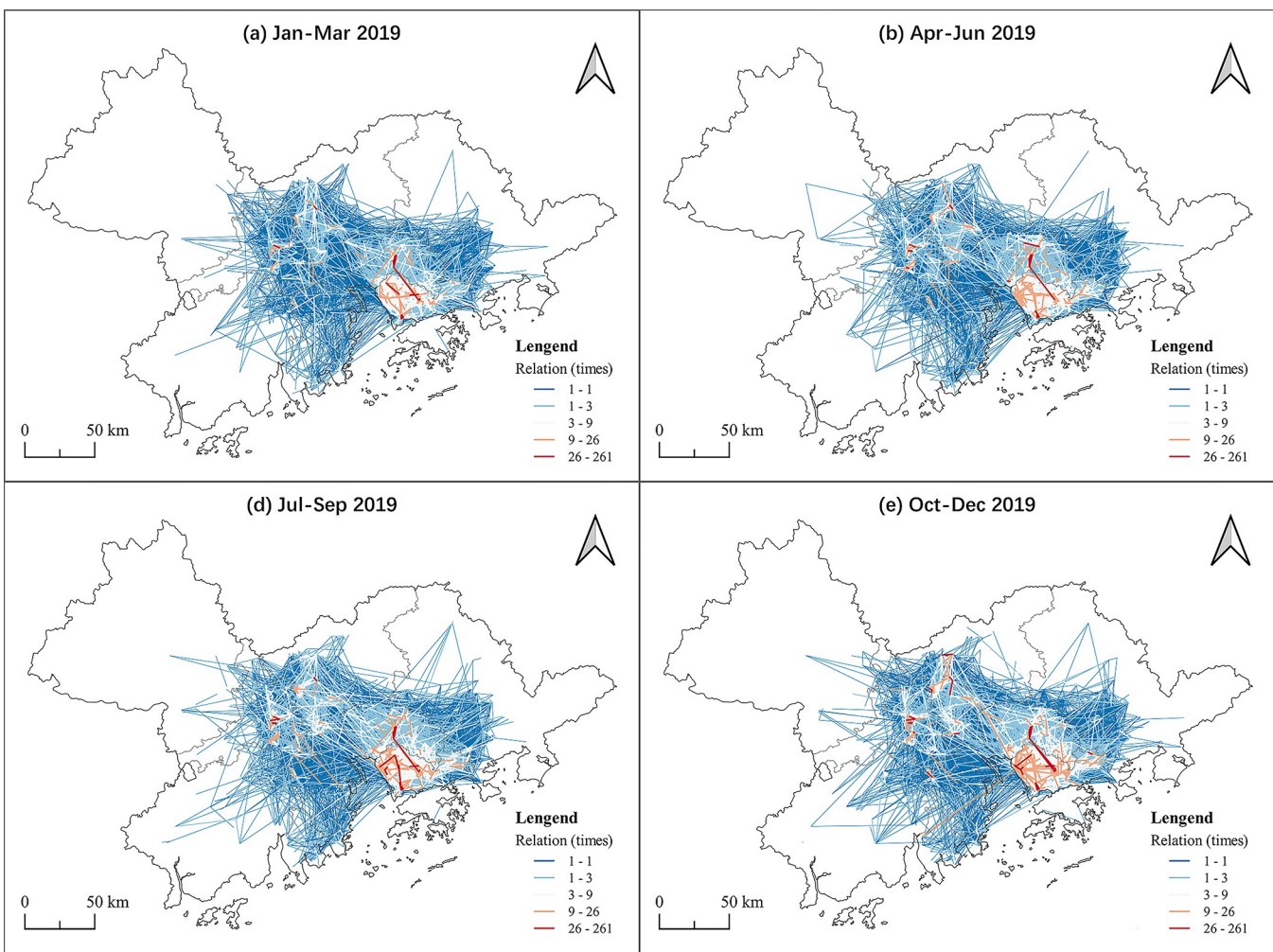

**Fig 3. The OD pairs at the micro-level in the GBA in 2019.** Reprinted background map from the National Catalogue Service for Geographic Information ([www.webmap.cn](www.webmap.cn)) under a CC BY license, with permission from the Ministry of Natural Resources of China, original copyright 2020.

Shenzhen. although e-commerce industrial parks and biochemical and agricultural ecological industrial parks mainly occurs within Zhongshan and Dongguan, and Zhongshan respectively. Some connections also occur within cities, such as the connection of e-commerce in Zhongshan and the biochemical and agricultural ecological industrial parks in Zhaoqing, and more across city connections should be strengthened to make the industrial parks of these cities be more closely connected with the GBA industrial network.

## The coupling relationship between industrial cooperation and industrial development

We have found that the industrial cooperation has been well constructed according to analysis of the industrial network based on multi-level OD pairs. In addition, the GBA has formed a clear division of labor in different cities, which leads to different cities playing various significant roles in the cooperative network. However, understanding the current industrial cooperation situation of the GBA is not enough to reveal the relationship of cooperative strategy and industrial development over a long period or to answer the question of whether the industrial

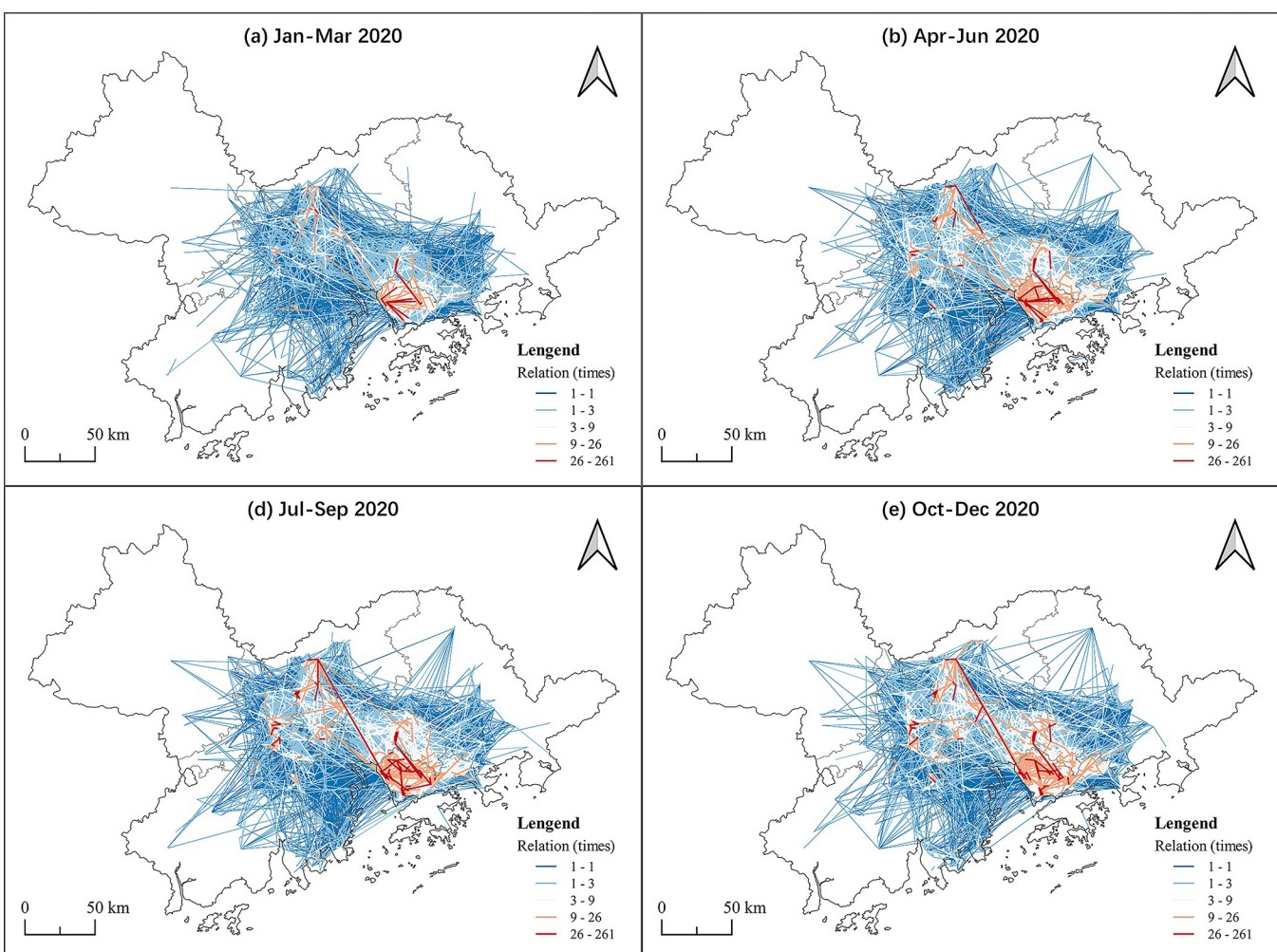

**Fig 4. The OD pairs at the micro-level in the GBA in 2020.** Reprinted background map from the National Catalogue Service for Geographic Information (www.webmap.cn) under a CC BY license, with permission from the Ministry of Natural Resources of China, original copyright 2020.

cooperation is necessary in regional economic development. Firstly, when dealing with the couple relationship, the coupling coordination model was introduced. It can be seen that the coupling coordination degree of all the cities has been stably increasing from 2010 to 2020 (see Fig 7 and Table 6). In the year 2010 among the nine cities only Jiangmen achieved primary coordination and the coupling coordination degree of most of the cities, such as Shenzhen, Zhuhai and Guangzhou, was only three and lay in moderate maladjustment. Almost all cities achieved good and high quality coordination at the end of 2020. The coordination value of Huizhou was only 0.65 and it achieved primary coordination at the end of 2020, similar to Zhongshan (0.87), Jiangmen (0.81) and Zhaoqing (0.80), although, with 9 coupling coordination grading, their coordination values are lower than other cities (such as Shenzhen and Dongguan) located in the more essential positions in the industrial networks constructed above. Considering the industrial collaboration networks in 2019 and 2020 discussed above and the coupling coordination degree, cities having stronger industrial cooperation leads to a higher coupling degree between industrial collaboration and industrial development.

Secondly, in terms of input efficiency, the industrial collaboration inputs (including collaborative sharing, collaborative learning and collaborative matching) have obtained the

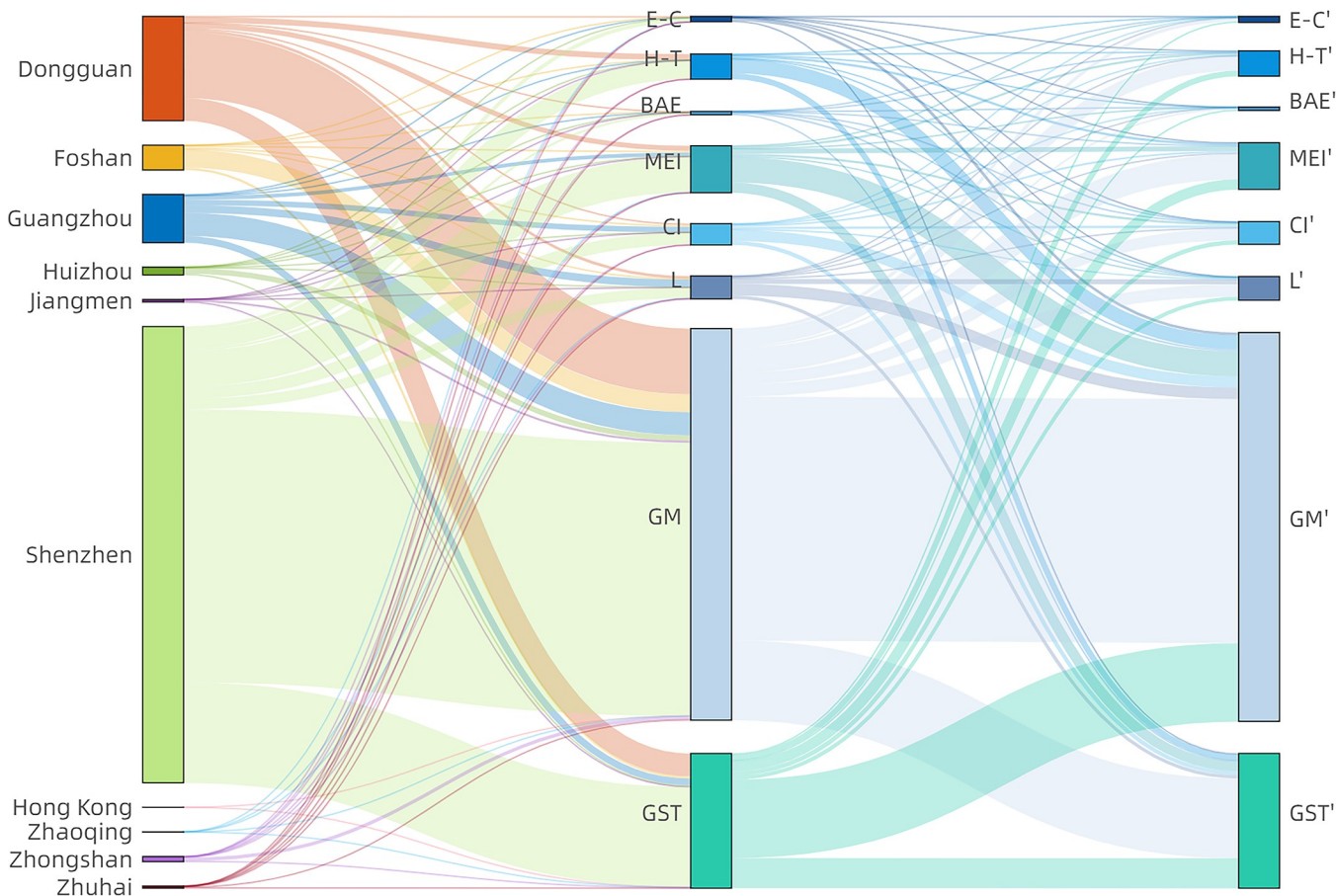

**Fig 5. The industrial cooperation of different types of industries in the GBA 2020.** E-C is short for e-commerce industrial park, H-T is short for high-tech park, BAE is short for biochemical and agricultural ecological park, MEI is short for mass entrepreneurship and innovation park, CL is short for cultural industry park, L is short for logistics park, GM is short for general manufacturing park and GST is short for General science and technology park.

economic development efficient (including industrial output, industrial environment and industrial efficiency) in each city of the GBA. However, there is also a small amount of investment redundancy (Fig 8). For example, Dongguan played a significant role in the construction of the industrial collaborative network, but its industrial development experienced cooperation input redundancy in 2014 and 2015. Zhuhai and Jiangmen also experienced a small amount of redundancy in 2012 and 2013, respectively, and this may have been caused by the strengthening of early industrial cooperation ties and the mismatched incomplete overall economic development level of the GBA.

## Discussion and conclusion

The study of industrial collaboration has a long history, ranging from early research on cooperation in equipment, steam, electricity, and other devices [67] to later investigations on the relationship between workers and organizational structures [18]. Subsequent studies gradually expanded to examine how cooperation and competition enhance the competitiveness of businesses and cities [17, 68]. Additionally, the research scope has evolved from focusing solely on industrial collaboration to encompass regional and international collaboration. Currently, industrial collaboration has become an important means of regional and intercity management. Economic instability, according to Hikmat [69], is unavoidable, unbalanced (full

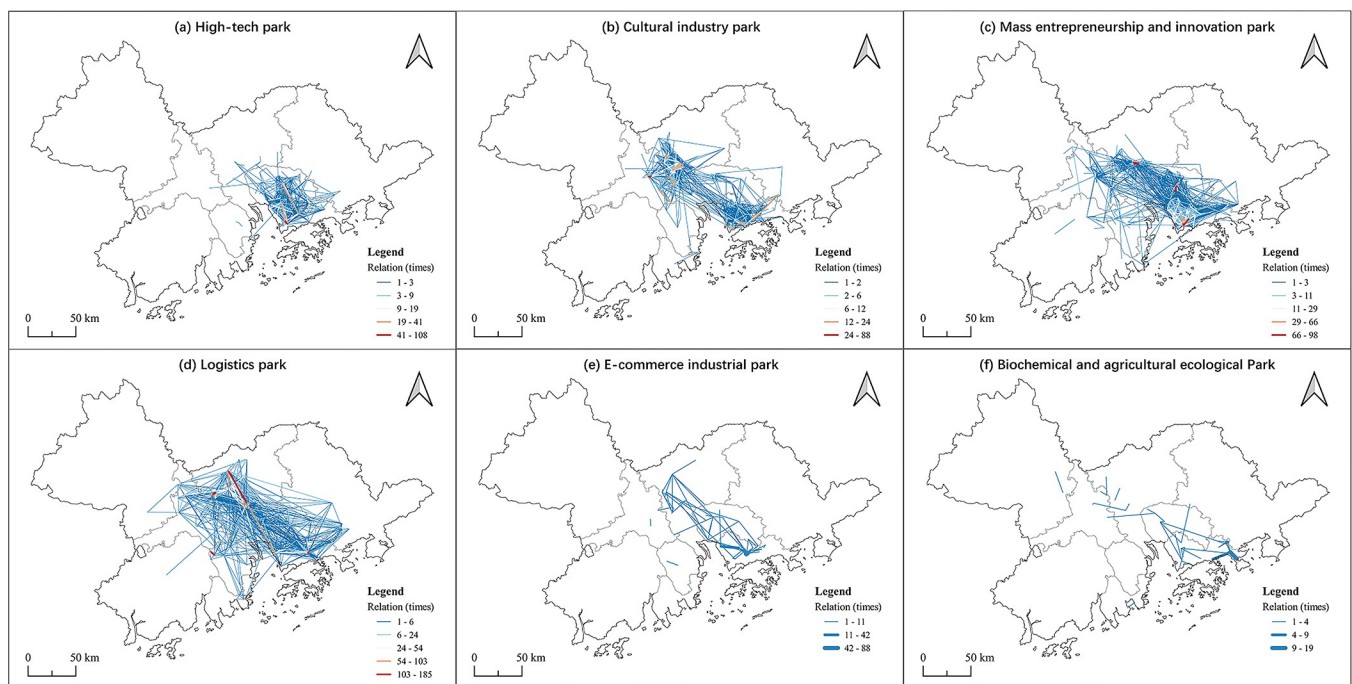

**Fig 6. The network of different industrial parks based on the OD pairs in 2019 and 2020.** Reprinted background map from the National Catalogue Service for Geographic Information (www.webmap.cn) under a CC BY license, with permission from the Ministry of Natural Resources of China, original copyright 2020.

employment), and requires government action [20]. In China, economic cooperation often serves as the main motivation for city governments to promote intercity collaboration [31, 70]. In China's urban regionalization process, market-driven economic connections, such as industrial cooperation and cross-border investments, appear to drive subsequent intercity collaboration in various thematic areas [71]. Research on industrial collaboration often employs case

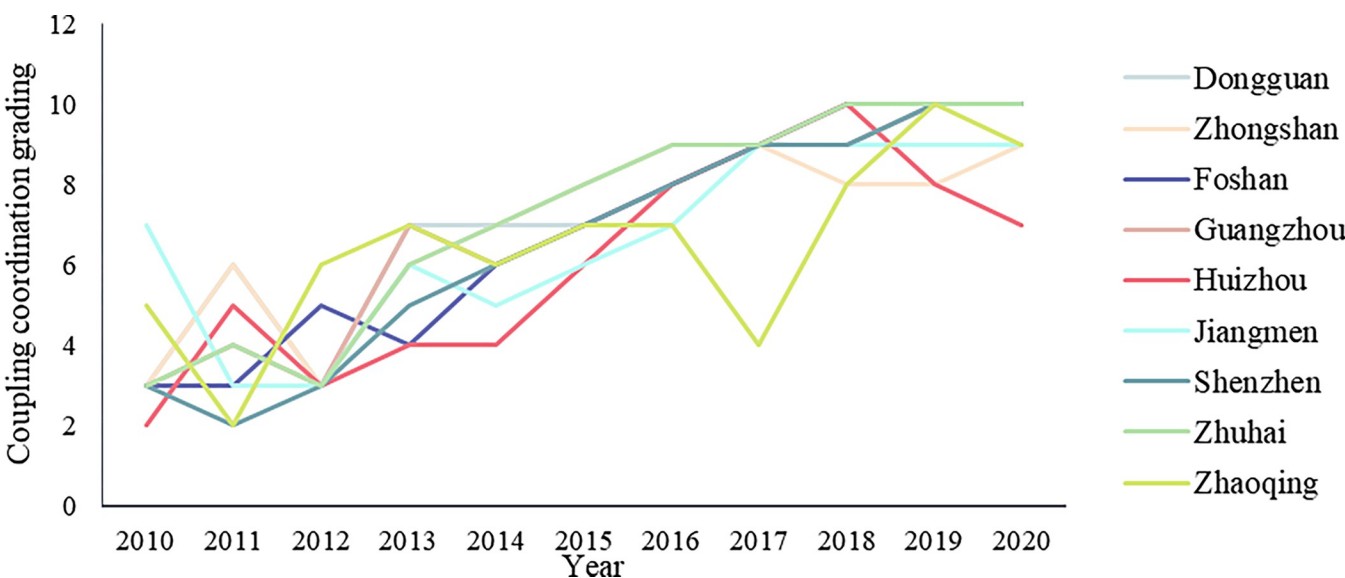

**Fig 7. The coupling coordination grading and degree of GBA cities.**

**Table 6. Meaning of coupling coordination grading.**

| Coupling coordination grading | Coupling coordination degree |
|---|---|
| 1 | Extreme maladjustment |
| 2 | Severe maladjustment |
| 3 | Moderate maladjustment |
| 4 | Mild maladjustment |
| 5 | On the verge of maladjustment |
| 6 | Grudging coordination |
| 7 | Primary coordination |
| 8 | Intermediate coordination |
| 9 | Good coordination |
| 10 | High quality coordination |

studies to explore cooperation and competition strategies across different periods [17], or utilizes interviews [19], lacking quantitative research methods and a comprehensive analysis of the impact of industrial collaboration on regional economic development.

Ottati [72] conducted the first study on the cooperation of industries within industrial clusters, suggesting that cooperation between industries can increase opportunities for large-scale activities among enterprises within industrial zones. Decision makers are inclined to establish industrial parks to provide a conducive environment for enterprise operations, leveraging the advantages of specialized labor markets, interrelated enterprises, and knowledge spillover

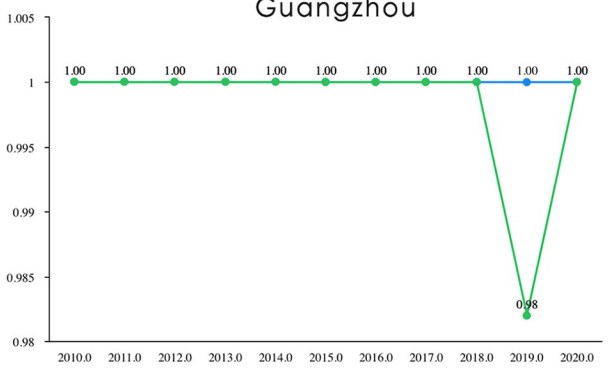
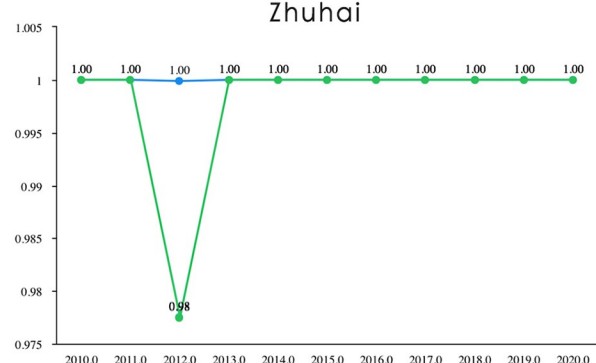
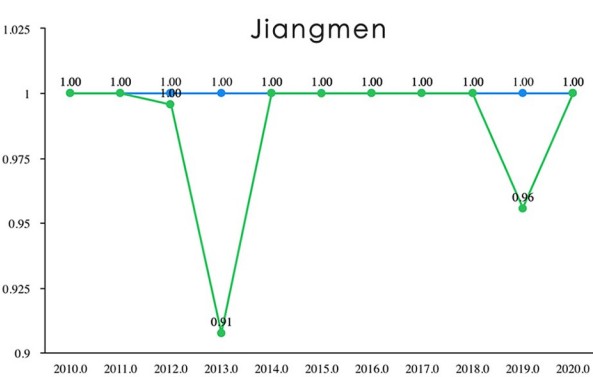
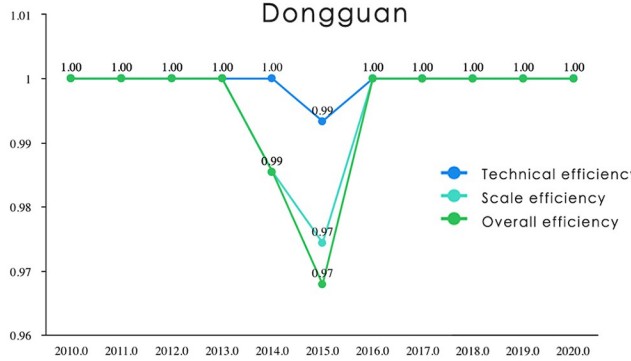

**Fig 8. The GBA cities with inefficient industrial cooperation inputs.**

effects [73]. Industrial parks, with their rapid growth worldwide, are often considered a driving force for regional development [24–26] and create specialized labor markets that support the production and operation of similar enterprises, leading to the agglomeration of similar businesses and the formation of connections between upstream and downstream enterprises, resulting in economies of scale [27]. The interconnection between industrial parks is also regarded as an important tool to assess regional cooperation [30–33]. The business world is composed of a network of interdependent relationships fostered through strategic cooperation agreements aimed at mutual benefits [74]. In the Greater Bay Area, there are over 8,920 registered industrial parks. However, due to the difficulty in establishing inter-regional industrial park networks, limited research focuses on the physical spatial connections between industrial parks and the spatial characteristics of industrial connection intensity.

Building upon this foundation, our study first utilizes big data to construct a measurement method for industrial collaboration based on inter-industrial park connections and measures the level of industrial collaboration and discussing its coupling relationship with economic development. In terms of intercity industrial collaboration, we found that the connections between Shenzhen and Dongguan, as well as Guangdong and Foshan, are significantly stronger compared to other regions. This is consistent with the concept of the "twin city strategy" [75] and "city-helps-city schemes" [76]. Intercity industrial cooperation has been widely adopted by governments at different levels in China as a panacea for addressing cross-regional governance challenges, and it is believed that the operational and effectiveness differences between these schemes are not substantial [70, 76]. However, practical outcomes have shown variations in terms of their effects [77]. Furthermore, as highlighted by Zhang et al [77], the level of geographic proximity plays a crucial role in determining the potential demand triggered by cooperation, thus increasing the intensity of collaboration. Higher levels of geopolitical proximity not only influence the intensity of cooperation in political agendas but also have implications for social and economic status. Our research reveals that cities with closer geographic proximity exhibit stronger cooperation, leading to better economic enhancement. In addition, Zeng et al [22] have proposed that inter-firm cooperation has a significant positive impact on the innovation performance of small and medium-sized enterprises (SMEs), while the connection and collaboration with government institutions do not have a significant effect on the innovation performance of SMEs. This finding is also applicable to the industrial collaboration in the GBA. In the GBA, Guangzhou, as the political and cultural center, lags behind Shenzhen, which serves as the economic and innovation hub in terms of cooperative development. In practice, it provides valuable information for policymakers and stakeholders to enhance their understanding of the industrial collaboration network in the GBA, fostering innovation and connectivity among enterprises. Additionally, the findings suggest that the Covid-19 pandemic has not had a significant impact on cross-city industrial cooperation within the urban agglomeration. This provides insights into how industries have adapted to the pandemic and highlights the importance of maintaining industrial collaboration during times of crisis. In fact, in regions with closer connections to other areas, the economic development has not been affected by the pandemic; instead, the connections between cities have become even stronger.

Duranton and Puga [78] identified three key aspects of the micro mechanism of regional economy: sharing, matching, and learning. Sharing refers to cost savings, increased diversified demands, and reduced risks. Moreover, as the number of units involved in matching increases, the effect becomes more significant at relatively lower costs. The spillover effect of knowledge is also crucial in the urbanization economy, as diversified knowledge contributes to the creation of new knowledge. Based on these foundations, the study developed an index system to measure the coupling relationship between industrial collaboration and industrial

development. Lai et al [23] revealed that industrial collaboration does not necessarily lead to industrial development. However, this study found that industrial collaboration and industrial development are dynamically coupled. With the enhancement of industrial collaboration systems and policies, the synergy and coordination between industrial development and collaboration gradually improve. On the other hand, the lack of industrial connections may result in lower levels of economic development. It is important to note that cooperative strategies are not a substitute for coordination and competition. Instead, they provide firms with additional options to compete more effectively [17]. Some cities, such as Zhaoqing and Huizhou, have received more benefits from industrial cooperation in the GBA than they have achieved individually. The research also suggests that taking industrial collaboration as an input variable effectively leads to industrial development outputs in almost every city in the GBA, highlighting the potential benefits of continued investment in industrial collaboration for economic development.

We can have some conclusion:

1. The industrial collaboration network has basically been formed, with the major cities and their surrounding satellite cities as the main representatives, presented particularly by Foshan and Guangzhou, and Shenzhen and Dongguan. The surrounding areas with backward economic development have gradually joined the network and have started to connect with other cities. Compared with the number of industrial park connections within the city, Zhaoqing and Huizhou have more connections in the urban agglomeration, and they may receive more benefits by being in the industrial collaboration network than relying on development of their own internal industrial connections.

2. The division of labor between the two major central cities is relatively clear. As a cultural and political center Guangzhou is inferior to Shenzhen as an innovation center in promoting the industrial connection between industrial parks. Guangzhou should take more responsibility for industrial transfer and for making supporting policies, such as a labor force export, to encourage more cities to integrate into the industrial synergy of the region, especially cities located in the west and southwest of the GBA.

3. The Covid-19 pandemic has not had a significant impact on the cross city level industrial cooperation of the urban agglomeration. Compared with 2019, in 2020, both annual and semi annual, the industrial links between cities was stronger, particularly with the areas with relatively backward economies playing stronger roles in the network. However, at the city level, the OD connections among cities were greatly impacted by the pandemic in the first quarter of 2019 and the number of connections in the city was significantly less than in the same period in 2020, although the barriers to traffic between cities was greater than it was within cities. This may be because some connections between industrial parks within cities can be avoided or replaced with non face to face communication and the flow of industries between cities cannot be easily substituted.

4. The degree of coupling between urban industrial coordination and urban industrial development has improved significantly in the recent 10 years, from moderate maladjustment to being basically well coordinated. Combined with the analysis of the industrial coordination network based on the OD pairs in 2019 and 2020 the cities with better industrial coordination have a higher degree of coupling of industrial coordination and economic development. That is a stronger industrial coordination leads to a more coordinated industrial coordination and industrial and industrial development. Similarly, the research found that taking industrial collaboration as the input variable, industrial collaboration efficiently leads to industrial development outputs in almost every city in the GBA. However, there is also a small amount of investment redundancy.

The practical implication of this study is to provide insights into the benefits and challenges of industrial cooperation in the context of economic uncertainties, such as those caused by the Covid-19 pandemic. By examining the industrial collaboration network in the GBA, the study provides valuable information for policymakers and stakeholders to enhance their understanding of the region's economic development and improve strategic planning and policy-making. The study also highlights the importance of regional coordinated development as a strategic priority for promoting steady economic growth and addressing economic uncertainties in developing countries.

The connections of industrial parks mainly came from 995893 OD pairs among 8920 industrial parks, which are based on AMaps App. Although AMaps is China's largest provider of navigation and real-time traffic information, other travel apps are also used in GBA so the limitations of data obtained cannot be avoid. In addition, because of the administrative and geographic boundaries, there is no sufficient data to form the connections between Hong Kong and Macao and other nine GBA cities. So in this paper, the evaluation of industrial cooperation of the GBA is based on nine cities in Chinese mainland.

## Acknowledgments

The authors gratefully thank the AMaps App travel OD data provided by AutoNavi Software Co., Ltd.

## Author Contributions

**Conceptualization:** Sa Ma, Jinge Ding.

**Formal analysis:** Sa Ma, Jinge Ding.

**Methodology:** Sa Ma, Jinge Ding.

**Visualization:** Sa Ma, Jinge Ding.

**Writing – original draft:** Sa Ma, Jinge Ding.

**Writing – review & editing:** Zhengdong Huang, Renzhong Guo.

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
