## [Decision Letter · Decision Letter 0]

27 Mar 2023

PONE-D-23-04060Evaluation of the industrial cooperation of the Guangdong Hong Kong Macao Greater Bay Area: based on the Origin-Destination pairs of industrial parks and coupling model efficiencyPLOS ONE

Dear Dr. Huang,

Thank you for submitting your manuscript to PLOS ONE. After careful consideration, we feel that it has merit but does not fully meet PLOS ONE’s publication criteria as it currently stands. Therefore, we invite you to submit a revised version of the manuscript that addresses the points raised during the review process.

We look forward to receiving your revised manuscript.

Kind regards,

Muazzam Sabir, Ph.D.

Academic Editor

PLOS ONE

Journal Requirements:

 "This research was support by the National Key Research and Development Program: (Project No: 2019YFB2103100); the Guangdong Science and Technology Strategic Innovation Fund (the Guangdong–Hong Kong-Macau Joint Laboratory Program, Project No.: 2020B1212030009); Shenzhen Key Laboratory of Digital Twin Technologies for Cities (Project No.: ZDSYS20210623101800001)." 

5. We note that Figures 2,3 and 4 in your submission contain [map/satellite] images which may be copyrighted. All PLOS content is published under the Creative Commons Attribution License (CC BY 4.0), which means that the manuscript, images, and Supporting Information files will be freely available online, and any third party is permitted to access, download, copy, distribute, and use these materials in any way, even commercially, with proper attribution. For these reasons, we cannot publish previously copyrighted maps or satellite images created using proprietary data, such as Google software (Google Maps, Street View, and Earth). For more information, see our copyright guidelines: http://journals.plos.org/plosone/s/licenses-and-copyright.

a. You may seek permission from the original copyright holder of Figures 2,3 and 4 to publish the content specifically under the CC BY 4.0 license.  

Additional Editor Comments:

**Reviewer #1:**

The manuscript addresses a topic that is very important and relevant to the regional economics. The objectives are stated clearly. However, this manuscript is not ready for publication yet. There are some suggestions that the authors need to address before the manuscript can be considered for publication.

S1. Introduction must be modified and restructured. Paragraph seems logical Incoherent statements rather than coherent paragraphs with a clear logical progression. Introduction should be reasonably brief, and its paragraphs should be placed in a logical sequence, without any repetition.

S2. Please add operational definitions.

S3. Results should be discussed with existing literature.

S4. What is the practical implication of this study?

S5. I didn’t find a discussion section, discuss the novelty, contribution, and future implications briefly.

S6. The paper needs to be professionally copy-edited as it contains several unclear sentences and linguistic inaccuracies.

**Reviewer #2:** Paper is well written but few suggestions are mentioned below

1. Abstract is well written but without any recommendation based on the finding of the study.

2. Although author wrote the significance of the study but the Problem statement is not clear in the article.

Overall it is a nice effort and may be processed further after incorporating the above suggestions satisfactorily.

Reviewers' comments:

Reviewer's Responses to Questions

**Comments to the Author**

1. Is the manuscript technically sound, and do the data support the conclusions?

Reviewer #1: Yes

Reviewer #2: Yes

2. Has the statistical analysis been performed appropriately and rigorously? 

Reviewer #1: Yes

Reviewer #2: Yes

3. Have the authors made all data underlying the findings in their manuscript fully available?

Reviewer #1: Yes

Reviewer #2: Yes

4. Is the manuscript presented in an intelligible fashion and written in standard English?

Reviewer #1: Yes

Reviewer #2: Yes

5. Review Comments to the Author

Reviewer #1:

The manuscript addresses a topic that is very important and relevant to the regional economics. The objectives are stated clearly. However, this manuscript is not ready for publication yet. There are some suggestions that the authors need to address before the manuscript can be considered for publication.

S1. Introduction must be modified and restructured. Paragraph seems logical Incoherent statements rather than coherent paragraphs with a clear logical progression. Introduction should be reasonably brief, and its paragraphs should be placed in a logical sequence, without any repetition.

S2. Please add operational definitions.

S3. Results should be discussed with existing literature.

S4. What is the practical implication of this study?

S5. I didn’t find a discussion section, discuss the novelty, contribution, and future implications briefly.

S6. The paper needs to be professionally copy-edited as it contains several unclear sentences and linguistic inaccuracies.

Reviewer #2: Paper is well written but few suggestions are mentioned below

1. Abstract is well written but without any recommendation based on the finding of the study.

2. Although author wrote the significance of the study but the Problem statement is not clear in the article.

Overall it is a nice effort and may be processed further after incorporating the above suggestions satisfactorily.

6. PLOS authors have the option to publish the peer review history of their article (what does this mean?). If published, this will include your full peer review and any attached files.

---

## [Author Response · Author response to Decision Letter 0]

25 May 2023

Dear Editors and Reviewers:

Many thanks for the constructive comments and suggestions from you and all reviewers. We greatly cherish the opportunity to review our manuscript. According to these valuable comments, we revised the original version of the manuscript to highlight the motivation and clarify some confusing sentences. The revisions focused on three aspects: (1) strengthening the motivation and importance of the research in the abstract and introduction. (2) adding discussion based on the results and existing literature. (3) ensuring that the manuscript meets the journal's requirements. These changes will not influence the content and framework of the paper. And here we did not list the changes but marked in revised paper.

We tried our best to improve the manuscript, and prepared a point-to-point response to the comments. We hope this revision will respond well to these valuable suggestions and suggested comments. 

We appreciate for Editors/Reviewers’ warm work earnestly, and hope that the correction will meet with approval. Once again, thank you very much for your comments and suggestions. Please feel free to let us know if there are further comments and suggestions. Thank you.

Sincerely,

Authors

 

Responds to the Journal Requirements:

Response: According to these additional requirements, we have carefully checked and revised the manuscript.

Response: We have revised the manuscript according to PLOS ONE style templates and confirm that the latest version adheres to PLOS ONE’s style requirements. Please refer to the manuscript.

2. Please note that PLOS ONE has specific guidelines on code sharing for submissions in which author-generated code underpins the findings in the manuscript.

Response: In this study, the code used to calculate the OD pairs between cities was based on Amap navigation data and base maps of administrative districts. Due to confidentiality agreements, please contact the corresponding author for the relevant code and data.

 "This research was support by the National Key Research and Development Program: (Project No: 2019YFB2103100); the Guangdong Science and Technology Strategic Innovation Fund (the Guangdong–Hong Kong-Macau Joint Laboratory Program, Project No.: 2020B1212030009); Shenzhen Key Laboratory of Digital Twin Technologies for Cities (Project No.: ZDSYS20210623101800001)." 

Response: This research was indeed supported by the above funds, but only the funder of project No. 2019YFB2103100 purchased Amap navigation data. Other funders had no role in study design, data collection and analysis, decision to publish, or preparation of the manuscript. For the specific statement, please refer to our latest cover letter.

Response: The POI and AOI data of industrial parks are available from the Amap official interface (https://lbs.amap.com/api/webservice/guide/api/search). The city-level GDP data can be found on the special page on Statistics of the GBA (https://www.dsec.gov.mo/BayArea/zh-CN/#s5). The data for the coupling model came from China's economic and social big data research platform (https://data.cnki.net). Amap navigation data are available from the GitHub database (https://github.com/Ding74/PLOSONE-MANUSCRIPT-D2304060).

5. We note that Figures 2,3 and 4 in your submission contain [map/satellite] images which may be copyrighted.

Response: The base maps of administrative districts shown in Figures 2, 3, 4 and 6 are all from the public version (2021) of basic geographic information data with a scale of one millionth authorized by the Ministry of Natural Resources and provided by the National Catalogue Service for Geographic Information (www.webmap.cn), its approval number is 2556 GS (2016). For the specific statement, please refer to our latest cover letter.

Response: Following the comments of the reviewers, we restructured the introduction and discussed the result with the existing literature. So, some new references have been added to the latest manuscript. Additionally, we have corrected some citation errors related to the literature format. For the revisions, please refer to the revised manuscript with marked-up.

 

Responds to the reviewers' comments:

Reviewer #1:

The manuscript addresses a topic that is very important and relevant to the regional economics. The objectives are stated clearly. However, this manuscript is not ready for publication yet. There are some suggestions that the authors need to address before the manuscript can be considered for publication.

Response: Thank you very much for your valuable comments. We have revised the manuscript based on your suggestions.

S1. Introduction must be modified and restructured. Paragraph seems logical Incoherent statements rather than coherent paragraphs with a clear logical progression. Introduction should be reasonably brief, and its paragraphs should be placed in a logical sequence, without any repetition.

Response: Thank you for your advice, the introduction part was restructured and modified. The introduction section of the paper highlights the following key points: 

(1) The global economy is facing uncertainties and challenges, particularly after the outbreak of the Covid-19 pandemic, including trade protectionism, supply chain fragmentation, and sluggish global economic growth. 

(2) Industrial collaboration is considered crucial in modern business environments and is a central aspect of strategic management. 

(3) In China, regional coordinated development has become a strategic priority for promoting steady economic growth and addressing economic uncertainties. 

(4) Insufficient research has been conducted on the coupling relationship between industrial cooperation and intercity economic development. 

(5) The paper focuses on the case study of the Guangdong Hong Kong Macao Greater Bay Area, utilizing extensive travel navigation big data to develop a method for evaluating industrial cooperation based on Origin-Destination (OD) pairs. The study also examines the non-spatial elements of industrial collaboration and their influence on industrial development. 

(6) The objective is to update research methods and gain a fundamental understanding of how industrial collaboration influences industrial development.

In summary, this introduction section presents the global economic challenges, emphasizes the significance of industrial collaboration, particularly in the context of regional development in China, and highlights the research gap regarding the evaluation methods of industrial cooperation and coupling relationship between industrial cooperation and intercity economic development. The study focuses on the Guangdong Hong Kong Macao Greater Bay Area as a case study, aiming to develop evaluation methods and enhance the understanding of the impact of industrial collaboration on industrial development. For the details, please refer to the revised manuscript, Introduction section.

S2. Please add operational definitions.

Response: We added the definition of industrial cooperation as follows: 

Industrial cooperation refers to the complementary and collaborative use of resources, information, and technology among different enterprises and industries within different geographic regions to achieve more efficient and competitive production and innovation. For the revision, please refer to the revised manuscript sentence 3, paragraph 2, Introduction section.

S3. Results should be discussed with existing literature.

Response: We have incorporated the existing literature and added a discussion section. For the details, please refer to the revised manuscript, Discussion and conclusion section. 

S4. What is the practical implication of this study?

Response: We have included the meaning of practical implication in the introduction, discussion, and conclusion sections. In summary, the practical implication including:

The practical implication of this study is to provide insights into the benefits and challenges of industrial cooperation in the context of economic uncertainties, such as those caused by the Covid-19 pandemic. By examining the industrial collaboration network in the GBA, the study provides valuable information for policymakers and stakeholders to enhance their understanding of the region's economic development and improve strategic planning and policy-making. The study also highlights the importance of regional coordinated development as a strategic priority for promoting steady economic growth and addressing economic uncertainties in developing countries.

S5. I didn’t find a discussion section, discuss the novelty, contribution, and future implications briefly.

Response: Similar to Question S3, we have supplemented the discussion section. For the details, please refer to the revised manuscript, Discussion and conclusion section.

S6. The paper needs to be professionally copy-edited as it contains several unclear sentences and linguistic inaccuracies.

Response: Thanks for your suggestions, we have proofread the article and made revisions to address grammar errors and unclear sentences. 

Reviewer #2: 

Paper is well written but few suggestions are mentioned below. Overall, it is a nice effort and may be processed further after incorporating the above suggestions satisfactorily.

Response: Special thanks to you for your precious comments. We have revised the manuscript based on your suggestions.

1. Abstract is well written but without any recommendation based on the finding of the study.

Response: We appreciate your suggestions, and based on the findings of our research, we have added some recommendations. They are mainly as follows:

Practically, decision makers should encourage and support intercity industrial collaboration, particularly between cities with closer geographic proximity, as it has been found to result in stronger cooperation and better economic enhancement. In addition, although industrial collaboration does not guarantee industrial development, when the collaboration systems and policies are enhanced, the synergy and coordination between them gradually improve. This highlights the potential benefits of continued investment in industrial collaboration for economic development.

2. Although author wrote the significance of the study but the Problem statement is not clear in the article.

Response: In the introduction and discussion part, we have emphasized the research problem. The problem addressed in the research is the study of industrial collaboration and its impact on economic development. It aims to understand the patterns of industrial collaboration within the GBA and examine the coupling relationship between industrial collaboration and economic development. The research also addresses the lack of quantitative methods and comprehensive analysis in previous studies, which focused primarily on case studies and interviews. For the details, please refer to the revised manuscript.

---

## [Decision Letter · Decision Letter 1]

10 Aug 2023

Evaluation of the industrial cooperation of the Guangdong Hong Kong Macao Greater Bay Area: based on the Origin-Destination pairs of industrial parks and coupling model efficiency

PONE-D-23-04060R1

Dear Zhengdong Haung,

We’re pleased to inform you that your manuscript has been judged scientifically suitable for publication and will be formally accepted for publication once it meets all outstanding technical requirements.

Kind regards,

Muazzam Sabir, Ph.D.

Academic Editor

PLOS ONE

Additional Editor Comments (optional):

Reviewers' comments:

Reviewer's Responses to Questions

**Comments to the Author**

1. If the authors have adequately addressed your comments raised in a previous round of review and you feel that this manuscript is now acceptable for publication, you may indicate that here to bypass the “Comments to the Author” section, enter your conflict of interest statement in the “Confidential to Editor” section, and submit your "Accept" recommendation.

Reviewer #1: All comments have been addressed

2. Is the manuscript technically sound, and do the data support the conclusions?

Reviewer #1: Yes

3. Has the statistical analysis been performed appropriately and rigorously? 

Reviewer #1: Yes

4. Have the authors made all data underlying the findings in their manuscript fully available?

Reviewer #1: Yes

5. Is the manuscript presented in an intelligible fashion and written in standard English?

Reviewer #1: Yes

6. Review Comments to the Author

Reviewer #1: (No Response)

7. PLOS authors have the option to publish the peer review history of their article (what does this mean?). If published, this will include your full peer review and any attached files.

Reviewer #1:

---

## [Editor Report · Acceptance letter]

16 Aug 2023

PONE-D-23-04060R1 

Evaluation of the industrial cooperation of the Guangdong Hong Kong Macao Greater Bay Area: based on the Origin-Destination pairs of industrial parks and coupling model efficiency 

Dear Dr. Huang:

I'm pleased to inform you that your manuscript has been deemed suitable for publication in PLOS ONE. Congratulations! Your manuscript is now with our production department. 

Kind regards, 

on behalf of

Dr. Muazzam Sabir 

Academic Editor

PLOS ONE